# Heat Acclimatization, Cooling Strategies, and Hydration during an Ultra-Trail in Warm and Humid Conditions

**DOI:** 10.3390/nu13041085

**Published:** 2021-03-26

**Authors:** Nicolas Bouscaren, Robin Faricier, Guillaume Y. Millet, Sébastien Racinais

**Affiliations:** 1Inserm CIC1410, CHU Réunion, 97448 Saint Pierre, France; 2Inter-University Laboratory of Human Movement Biology, UJM-Saint-Etienne, Univ Lyon, EA 7424, 42023 Saint-Etienne, France; robinfaricier@live.fr (R.F.); guillaume.millet@univ-st-etienne.fr (G.Y.M.); 3Institut Universitaire de France (IUF), 75231 Paris, France; 4Research and Scientific Support Department, Aspetar Orthopedic and Sports Medicine Hospital, Doha 29222, Qatar; Sebastien.Racinais@aspetar.com

**Keywords:** hot temperature, hydration, dehydration, electrolyte balance, body temperature regulation, acclimatization, ultra-endurance running, running

## Abstract

The aim of this study was to assess the history of exertional heat illness (EHI), heat preparation, cooling strategies, heat related symptoms, and hydration during an ultra-endurance running event in a warm and humid environment. This survey-based study was open to all people who participated in one of the three ultra-endurance races of the Grand Raid de la Réunion. Ambient temperature and relative humidity were 18.6 ± 5.7 °C (max = 29.7 °C) and 74 ± 17%, respectively. A total of 3317 runners (56% of the total eligible population) participated in the study. Overall, 78% of the runners declared a history of heat-related symptoms while training or competing, and 1.9% reported a previous diagnosis of EHI. Only 24.3% of study participants living in temperate climates declared having trained in the heat before the races, and 45.1% of all respondents reported a cooling strategy during the races. Three quarter of all participants declared a hydration strategy. The planned hydration volume was 663 ± 240 mL/h. Fifty-nine percent of the runners had enriched their food or drink with sodium during the race. The present study shows that ultra-endurance runners have a wide variability of hydration and heat preparation strategies. Understandings of heat stress repercussions in ultra-endurance running need to be improved by specific field research.

## 1. Introduction

Hot and humid ambient conditions impair prolonged exercise capacity and may favor exertional heat illness (EHI) [1,2]. Its most severe form, exertional heat stroke (EHS), is characterized by a core body temperature above 40.5 °C associated with central nervous system dysfunction (delirium, convulsions, coma) [3,4]. EHS is considered the second most common cause of sport- and exercise-related sudden death, after cardiac conditions [5]. Thus, several guidelines regarding training and competing in the heat have been published [6,7]. These guidelines are mostly based on laboratory studies [6] and are limited to running events up to the marathon distance. In other words, they may not apply to ultra-endurance sport, defined in the present paper as any event longer than 6 h [8,9]. In ultra-endurance sports, the thermoregulation and hydration challenges are different compared to shorter exercises due to the moderate intensity but prolonged duration, sometimes associated with exotic destinations, hot or warm environments (desert, tropical climate), and with high prevalence of digestive disorders and pathological processes in various organs such as skeletal muscles, heart, kidneys, and immune and endocrine systems [9,10,11]. Because of these specificities, the transposition of current knowledge and guidelines for road-races up to marathon toward ultra-endurance running is hazardous [9,10,12].

With the growing popularity of ultra-endurance events, attention to the unique needs of this population becomes increasingly important [13]. Only a few studies have focused on the influence of heat on performance [14], exercise-associated hyponatremia [15,16], or hydration requirements [17,18,19] in ultra-endurance events. Areas of uncertainty remain regarding thermoregulatory function, EHI prevalence, and health impairment during ultra-endurance running events in warm or hot conditions [1,13,19]. Moreover, despite the well-accepted benefits of heat acclimatization and cooling strategies before running in hot and humid conditions [6,20], it remains unknown whether ultra-endurance runners follow those guidelines from classical endurance activities.

Moreover, debates continue as to how endurance athletes should hydrate during exercise, i.e., ad libitum/drinking to thirst vs. pre-programmed drinking schedule [21]. For Kenefick et al. [21], drinking to thirst is more applicable for short-duration endurance exercise (<90 min), whereas a programmed drinking strategy should be tailored to prevent body mass losses or gains of ± 2% during activities longer than 2 h in duration, especially if they take place in warm or hot environments. For Hoffman et al. [12], hydration guidelines suitable for short periods of exercise in classical endurance events are not appropriate for ultra-endurance activities and may even cause harm. Hyperhydration can potentially cause the serious complication of exercise associated hyponatremia, so ultra-endurance athletes have been advised to avoid drinking beyond the dictate to thirst [12]. In addition, while the sodium consumed with a typical race diet appears to be adequate to replace losses and avoid salt-depletion in ultra-endurance [22,23], hydration guidelines recommend ingestion of sodium during endurance exercise, and use of supplementation has been a common practice among ultra-endurance athletes [24].

Given the lack of information about heat mitigation strategies in ultra-endurance runners before a race in a tropical environment and the controversies about hydration and sodium supplementation during endurance exercise, the aim of the current study was to survey heat preparation, cooling strategies, and heat-related symptoms (HRS) and to focus on hydration and sodium intake during an ultra-endurance running event in warm and humid environment.

## 2. Materials and Methods

### 2.1. Study Design

This survey-based cross-sectional study was open to all people who participated in one of the three races of the 2019 edition of the Grand Raid de la Réunion. A survey was distributed before and after the race. E-versions were transmitted by the organizing committee through a mailing-list. A hard copy of the pre-race survey was also available at race-bib collection.

### 2.2. Ethics Approval

This study was approved by the Saint-Etienne University Hospital ethics committee (#IRBN 572019/CHUSTE) and registered in ClinicalTrials.gov (#NCT04136925). The data were collected, recorded, and stored after obtained consent from the research subjects and were included in the register of processing activities of the Réunion Island University Hospital Center.

### 2.3. Characteristics of the Races

The event was held in Réunion Island (a tropical island in the Indian Ocean) and included three races: *La Diagonale des Fous* (DDF; 2900 runners, 165 km, 9576 m of positive and negative elevation, 2019 finish time range (23:33:45–66:04:00)); *Le Trail de Bourbon* (TDB; 1600 runners, 111 km, 6433 m elevation, (15:34:56–41:54:16)); and *La Mascareignes* (MAS; 1700 runners, 65 km, 3505 m elevation, (07:43:55–20:23:19)). Environmental conditions measured across 8 weather stations distributed along the three running courses reported a temperature of 18.6 ± 5.7 °C (range: 3.6 to 29.7 °C), a relative humidity of 74 ± 17% (5 to 100%), a dew point of 13.0 ± 5.9 °C (−24.7 to 23.4 °C), and a solar radiation of 87 ± 116 °J/cm^2^ (0 to 402 °J/cm^2^).

### 2.4. Participants

A total of 3317 runners participated in the study, representing 56% of the total eligible population. Sixty-nine percent (*n* = 2286) of study participants completed the pre-race survey, and 62% (*n* = 2050) completed the post-race survey. Overall, 1763 respondents took part in the 165-km race, 882 in the 65-km race, and 665 in the 111-km race; 83% of the responders (both pre- and post-race survey) were finishers. The sex ratio was 5:1 for men, and the mean age of runners was 42.2 years (representative of the starters). The climate of residence was temperate or continental for 55% of runners, and hot (tropical or dry) for 45% of runners. The demographic, morphological, and training characteristics of the participants are presented in Table 1.

### 2.5. Survey

The pre-race survey (Appendix A) contained questions on demographic characteristics and on trail-running and training experience, followed by specific questions on trail-running experience in hot environments, medical history of heat symptoms and heat illness before this race, and heat acclimatization before the race, and then by questions on planned hydration and cooling strategies. Cooling strategy was defined as any method used for reducing or preventing excessive heat storage during exercise. Only mid-cooling (i.e., cooling during the races) strategies used by runners were assessed. To help the runners answering, the main cooling methods were provided in the questionnaire (Appendix A). The post-race survey (Appendix A) looked at the impact of environmental conditions on performance, the occurrence of HRS during the race, and questions on effective hydration during the race. Planned hydration was asked in pre-race survey by “How amount of fluid do you plan to drink during the race? (in mL/h)”, and effective hydration was asked in post-race survey by “What amount of fluid did you drink during the race (in mL/h)?” (Appendix A). Data on runners (i.e., withdrawal status and finish time) were provided by the organizing committee of the Grand Raid de la Réunion.

### 2.6. Statistical Analyses

Qualitative variables were expressed as numbers and percentages, and quantitative variables were expressed as medians (Q1–Q3) or means ± SD. Univariate analyses were performed using Pearson’s chi-squared test for categorical variables, and the Student’s t-test or the Mann–Whitney U test for quantitative variables as appropriate. The Kruskal–Wallis test and the Wilcoxon rank-sum test were performed to compare fluid volume consumption between the races. The significance level was set to 5%. Bonferroni correction was applied to multiple comparisons. All analyses were performed using Stata V13 software (StataCorp LP, College Station, TX, USA).

## 3. Results

### 3.1. Medical History of Heat Related Illness before This Race

Before this race, 78% of participants reported a history of HRS; this value was higher in men than women (79.9% vs. 70.2%, *p* < 0.001), and higher in runners living in hot climates (HCR) than runners living in temperate climates (TCR) (80.5% vs. 75.3%, *p* = 0.003). A higher prevalence of a history of muscle cramps was found in men compared to women and in HCR compared to TCR (*p* < 0.001) (Table 2). A total of 44 (1.9%) runners reported a previous diagnosis of EHI; of these, 18 (0.8%) had been hospitalized or had visited the emergency room and 4 (0.2%) had received intensive care. A higher prevalence of heat-related hospitalization was found in TCR compared to HCR (*p* = 0.021). No difference was found according to sex (Table 2).

### 3.2. Heat Training and Acclimatization

About one-quarter (24.3%) of the 1033 TCR declared having trained in the heat. This value was similar between men and women (21.8% vs. 24.8%, *p* = 0.427) independently across the three races (*p* = 0.599). Most of these runners (98.4%) used natural heat (length of preparation: 15 (10–30) days) while only five runners used a climatic chamber (length of preparation: 6 (3–9.5) days). Among TCR, 56.8% declared having voluntarily scheduled an early arrival on the island to acclimatize to local environmental conditions (outside a holiday context), a proportion that was higher in women (65.2%) than men (55.2%, *p* = 0.022). However, the time to start of the race was only 4 (3–6) days with 6.8% of runners landing the day before the start of the race and 86.7% landing within 7 days before the race. No significant difference in arrival times before the race was found between TCR finishers and TCR non-finishers (4 (3–6) vs. 5 (3–6), respectively, *p* = 0.163).

### 3.3. Cooling Strategy

A cooling strategy during the race was reported by 45.1% of runners with no difference according to sex (*p* = 0.107) or climate of residence (*p* = 0.569) (Table 3).

### 3.4. Heat Related Symptoms during the Races

The post-race survey revealed that 23.8% of runners suffered from the heat during the race; this value was similar according to sexes (*p* = 0.898) but higher in HCR than TCR (25.8% vs. 19.9%, *p* = 0.003). These findings were consistent across the three races. A similar proportion of runners (20.2%) thought that the heat had negatively impacted their performance, without reaching the level of significance between HCR and TCR (*p* = 0.078). Up to 54.6% of the runners reported at least one HRS; this value was higher in men than in women (56.5% vs. 46.8%, *p* = 0.002) but did not vary according to climate of residence (*p* = 0.603). Runners declared the following HRS: muscle cramping (43.4%), digestive disorder (14.3%), collapse (1.2%), and severe headache (0.9%). A detailed list of HRS according to sex is provided in Figure 1.

### 3.5. Hydration

The hydration rate planned by runners was 663 ± 240 mL/h (corresponding to 9.7 ± 3.7 mL/kg/h) for the whole population (686 ± 237 mL/h for MAS vs. 640 ± 235 mL/h for TDB vs. 661 ± 242 mL/h for DDF (*p* = 0.018)). This rate was 624 ± 229 mL/h for women vs. 671 ± 241 mL/h for men (*p* = 0.002) (see repartition per race in Figure 2A), and 635 ± 236 mL/h for non-finishers vs. 671 ± 240 mL/h for finishers (*p* = 0.009). No differences in planned hydration volume were found according to climate of residence (*p* = 0.569). More men than women planned to drink sodium-enriched water (33 vs. 24%, *p* < 0.001) (Table 3). The consumption of pure water or homemade preparation was more prevalent in HCR (*p* < 0.001), whereas the consumption of sodium-enriched water was more prevalent in TCR (*p* < 0.001). Ten percent of runners planned to drink only pure water during the races, namely 13.5% of MAS runners vs. 11.0% of TDB runners vs. 9.0% of DDF runners (*p* = 0.013). Repartition of beverage composition extrapolated on a 1-L bottle is illustrated in Figure 2C. Qualitative analysis of the variables “homemade preparation” and “other kind of beverage” found that 24.0% of runners planned to add carbohydrates to their beverage, 12.6% planned to consume sparkling beverages, and 11.9% planned to consume soda drinks.

Overall, 77.0% of runners declared having a hydration strategy; this value was higher in HCR than TCR (79.9% vs. 74.9%, *p* = 0.013), but did not vary according to sex (*p* = 0.518). Drinking to thirst was declared by 39.2% of runners. Other factors determining runners’ hydration were color of urines (10.9%), and maximum tolerated hydration (6.4%). The value of hydration volume reported post-race was 61 ± 244 mL/h lower than the value planned by the runners pre-race (*p* < 0.0001); this difference was consistent across the three races (Figure 2B). According to the post-race survey, 80.3% of runners considered their hydration to be sufficient; this value was lower in women than men (73.9% vs. 81.5%, respectively, *p* = 0.002) but did not vary according to climate of residence (*p* = 0.301). Moreover, 58.8% of the runners added sodium to their food or beverage during the race; this value was higher in HCR than TCR (62.9% vs. 53.8%, *p* < 0.001) but did not vary according to sex difference (*p* = 0.585). The proportion of runners (51.8%) who ate soup during the race was similar between HCR and TCR.

## 4. Discussion

The main results of the present study are that (i) although 78% of ultra-endurance runners had a previous history of HRS, 1.9% declared a medical history of EHI; (ii) only one quarter of TCR reported having specifically trained in the heat, yet the prevalence of self-declared negative impact of environmental conditions in performance (20%) and HRS incidence (54.6%) was not higher for TCR compared to HCR; (iii) three quarter of all participants had a hydration strategy, with thirst representing a hydration signal for 39% of them, and 59% of runners added sodium to their food or beverage during the race. To our knowledge, this is the first study describing the heat mitigation strategies in ultra-endurance runners before a race in a tropical environment.

### 4.1. History of Heat-Related Symptoms and Exertional Heat Illness

Three-quarters of participants reported a history of HRS before this race, which is higher than what has been reported by elite athletes (48%) and elite cyclists (57%) [25,26]. Conversely, forty-four runners (2%) of the athletes participating in the present study reported a history of EHI, leading to hospitalization or emergency consulting in 18 (0.8%) of them. This prevalence is lower than in previous cross-sectional study in elite athletes (16% in road cycling [26] and 9% in athletics [25]). This relatively low EHI prevalence in ultra-endurance running compared to other sports must be confirmed by prospective epidemiological studies but can partially be explained by the fact that exercise intensity is at least as important as environmental conditions for increasing core temperature [19,26]. Another explanation can be found in the “flush model” developed by Millet [27]. In his holistic model, the author suggests that elite athletes (who are very rare in ultra-trail pelotons) could finish the race with a lower security reserve (i.e., a reserve allowing one to prevent physiological damage) and therefore could be more exposed to EHI in hot and humid environments than amateur ultra-endurance runners [27]. Finally, a relative low prevalence of EHI compared to HRS declared by runners could be related to an under-diagnosis of the true rate of EHI [28]. Given the frequent failure to measure core body temperature during races, it is plausible that some events could be erroneously attributed to cardiac conditions based on incidental pathological findings, whereas heat stroke was the real etiology. A previous study suggested that serious cardiac events were outnumbered by heat stroke events by a factor of 10 during endurance sport [29].

### 4.2. Heat Training and Acclimatization

Heat acclimatization is considered as the most important countermeasure to protect the health of athletes and to enhance their performance in hot conditions [6,30]. Ideally, the heat acclimatization period should pass 2 weeks in order to maximize all benefits [6]. In our study, only one-quarter of TCR reported having trained in the heat (mostly in natural environments) before the event, with an average training duration of 15 days. This percentage is higher than that observed in the high-performance athletes who participated in the 2015 IAAF Worlds Athletics Championships held in Beijing (15%) [25]. The comparison with the Worlds Athletics Championships study must however be made with caution because this study included track and field athletes for which heat acclimatization is less important [6]. Considering the repercussions of heat stress on endurance performance [14,31,32], and given the fact that heat acclimatization can reduce the likelihood of heat illness [20], one may wonder why so few runners in our study acclimatized before the event. We suggest two possible explanations for this. First, this may be due a lack of awareness of the potential risks in this population due to the scarcity of epidemiological data on the prevalence of EHI in ultra-endurance running and to the absence of guidelines and recommendations for ultra-endurance runners [13]. Second, ultra-endurance runners are mostly amateurs with a limited possibility to organize training camps. In our study, 42% of the runners who had trained in the heat before the event reported a training duration less than 14 days. Half of our eligible population was made up of runners from metropolitan France, where the climate is temperate (mean temperature in October 2019 was 15.1 °C), making optimal heat-acclimatization difficult. The time required to achieve optimal acclimatization likely varies, but a total period of 2 weeks has been shown to facilitate maximal adaptations [6,33,34]. In our study, the time to start of the race was only 4 days. In the present study, no significant difference in arrival times before the race was found between TCR finishers and TCR non-finishers (*p* = 0.163).

### 4.3. Cooling Strategy

While several reviews concluded that cooling can increase prolonged exercise capacity in hot conditions [34,35], only 45% of ultra-endurance runners in our study used cooling strategies during the race. This proportion is lower than mid-cooling strategy using prevalence (98%) found in elite road race athletes during the Doha 2019 IAAF World Athletics Championships [36]. In our study, runners used mostly external cooling strategies, namely natural cooling strategies (leg (7%) or whole-body (4%) immersion in creeks or rivers, resting in the shade (19%)) and classic strategies (wet sponge (12%), head/neck cooling (17%), hat, cap (12%), cold towel (4%), neck collar (3%)).

### 4.4. Heat Related Symptoms during the Races

During the races, 24% of runners declared having suffered from the heat, and only 20% stated that the heat had negatively impacted their performance. This relative low prevalence of perceived heat repercussions may be partly due to the relatively mild conditions that prevailed during this edition of the Grand Raid de la Réunion, with temperatures not exceeding 30 °C (range 3.6 to 29.7 °C) and humidity remaining at usual levels (75%). This was in fact the coolest edition of the last 6 years (range 5.8–32.3 °C). Importantly, only 61 non-finishers answered the post-race survey (accounting for 3% of all completed post-race survey), which contained questions on HRS during the race. Considering that the proportion of non-finisher of the three races of the 2019 edition was 28%, heat repercussions during races were most probably underestimated. In future studies, data should be collected from the onsite medical team and from local hospitals to determine the prevalence of EHI during the race. Up to 55% of the runners reported at least one HRS, which is surprising given that only 20% of the runners stated that the heat had negatively impacted their performance. The reasons are unclear. This gap may be explained by the fact that respondents reporting symptoms were not due to the heat but to other issues (injuries, sleep deprivation, etc.), even though the survey question was explicitly about HRS. It is also possible that runners had symptoms but considered that it did not limit their performance (i.e., they considered to be limited by other factors). This may be further exacerbated by declarative bias, i.e., runners would appear strong and declare symptoms without mentioning it was influencing their performance. Although only a quarter of athletes of TCR reported having trained specifically in the heat, similar prevalence of negative impact of environmental conditions in performance (~20%) and HRS incidence (55%) during races were declared according to climate of residence. Moreover, a higher percentage of HCR declared having suffered from the heat (25.8% vs. 19.9%, *p* = 0.003). This unexpected difference could be partly explained by the date of the race and the importance of seasonal acclimatization. The race took place in mid-October, just after the end of summer for the inhabitants of the northern hemisphere, but just before the start of the hot season (November) for local residents (mean temperature in October 2019 in Réunion Island was 22 °C). Although temperatures in France were only 15 °C at the time of the race, the temperature in the month preceding the events could rise to above 25 °C, allowing the TCR runners to repeatedly train in warm environments. Residing in a hot climate may not confer an advantage if one has not been yet exposed to heat for many months, i.e., has not benefited from seasonal acclimatization of a hot season as was the case from the runners from La Réunion, i.e., the vast majority of our HCR population [5]. This is critical, as seasonal acclimatization is important in protecting athletes’ health when practicing in the heat. Indeed, it has been shown that the risk of heat related collapse was higher at the beginning than the middle of the summer when athletes had not yet acquired natural acclimatization to heat [37]. Moreover, a recent study of analysis of heat illness in the Beach Volleyball World Tour showed that there were more medical time-outs related to the heat during competition in Asia in the winter when northern hemisphere players do not benefit anymore from seasonal acclimation [5].

### 4.5. Hydration

Data on hydration strategies during ultra-endurance running remain scarce [13]. In our study, 77.0% of runners declared having a hydration strategy, with a planned hydration rate of 663 mL/h (corresponding to 9.7 ± 3.7 mL/kg/h), with a higher hydration rate in shorter races: 686 ± 237 mL/h for MAS (65 km) vs. 640 ± 235 mL/h for TDB (111 km) vs. 661 ± 242 mL/h for DDF (165 km) (*p* = 0.018). These results are in line with recent publications in ultra-endurance running reporting a mean hydration volume of 685 [38] and 732 mL/h [39]. Past consensus statements recommended minimizing fluid deficit; however, debates continue as to how athletes should hydrate during exercise, i.e., ad libitum vs. pre-programmed drinking schedule [18]. The typical hydration guidelines to avoid more than 2% body mass loss may not apply in ultra-endurance activities [12]. In our sample, thirst was the factor determining hydration for only 39% of runners; the other factors were color of urine for 11% and maximum tolerated hydration for 6% of the runners. Recent considerations of hydration concluded that ultra-endurance runners should be cautious to avoid drinking beyond the dictate of thirst and taking in excessive sodium during prolonged exercises [12]. Whilst hyperhydration can lead to exercise associated hyponatremia, hypohydration during exercise in hot ambient conditions can increase the risk of developing EHI [40]. Moreover, pronounced dehydration associated with influx of muscle protein (myoglobin) caused by muscle damage may lead to kidney damage. The prevalence of an acute kidney insult in ultra-marathon running is nearly 45% of all runners [41].

A wide range of hydration strategies was observed in our population. Most runners (75.3%) had planned to drink water; 59.8% had planned to consume exercise drinks, almost 60% had planned to consume sodium-enriched water or food, 24% had planned to consume carbohydrate-enriched water or food, and 10% of the runners declared drinking exclusively pure water during the race. Hydration and nutrition recommendations commonly prescribe sodium and carbohydrate ingestion during prolonged endurance exercise in the heat [42]. However, sodium supplementation should not be aimed at replacing all losses and should not be excessive during ultra-endurance activities, as sodium consumption in the typical race diet of ultra-endurance runners appears to be adequate [12].

In our study, the heterogeneity of hydration practices is reflected in the extreme diversity of products that were added to homemade preparations by runners: coffee or tea, fruit juice, spices, milk, plants or herbs, proteins, amino acid supplements, etc. Of note, the challenges of endurance sports include the potential for large variations in ambient conditions during a single event, and practical considerations include the availability of nutrition supplies at aid stations, the difficulties of ingestion during exercise, and the interaction with gastrointestinal comfort/function [42]. Specific recommendations are needed regarding hydration volume and drink content during ultra-endurance running adapted to (i) individual tastes and avoidance of loss of appetence, and (ii) the practical considerations associated with each race (autonomy or semi-autonomy).

### 4.6. Study Strengths and Limitations

The low rate of response to the post-race survey among non-finishers may have biased our analysis of the effects of hot and humid conditions on ultra-endurance running. As this study is a survey-based study, it is subject to the declarative bias of athletes. It remains possible that despite the specify of the questions regarding HRS and HRI, some runners attributed to heat some conditions being not heat-related. Comparisons of prevalence made with the prevalence of other disciplines or other papers should therefore be considered with caution due to the lack of clinical data in our study to confirm the self-assessment of the runners. However, our study provides a relevant estimation of the burden of EHI self-reported in ultra-endurance running, a discipline poorly investigated. Nevertheless, a major strength of the present study is its total sample size (*n* = 3317) and representativeness, as 56% of the total eligible population was included in at least one of the two surveys (pre- and post-race surveys). Environmental conditions during the race were not extreme due to relatively low temperatures; since runners did not know ahead of time what environmental conditions would be like, our findings concerning the strategies they employed (as declared on the pre-race survey) can be considered as representative of the population.

## 5. Conclusions

The present study shows that ultra-endurance runners have a wide variability of hydration and heat preparation strategies. However, although consensus recommendations on training and competing in the heat are only partially adopted by ultra-endurance runners, prevalence of EHI remains low in ultra-endurance, probably because exercise intensity is a more potent parameter for increasing body temperature than environmental parameters. Being native as a resident of tropical country seems not to confer an advantage in reducing the negative impact of heat on performance and HRS incidence if the athletes have not been exposed to heat in the weeks preceding the race (i.e., no seasonal acclimatization). In our study, 3/4 of runners declared having a hydration strategy, with a planned hydration volume of about 650 mL/h. Thirst represented the hydration signal for 40% of ultra-endurance runners, and 60% of them added sodium to their food or beverage. Recent considerations recommending that ultra-endurance runners should be cautioned to avoid drinking beyond the dictate of thirst and taking in excessive sodium during prolonged exercises are only partially respected. Understandings of heat stress repercussions in ultra-endurance running need to be improve by specific field research.

## 6. Practical Implications

Information of the ultra-endurance runners about benefits of heat acclimatization and cooling strategies before running in hot and humid conditions is needed.Prevalence of EHI remains low in ultra-endurance based on declarative evaluation in this study. However, prospective studies with clinical assessment of EHI (core temperature, symptoms) to better estimate the burden of heat stress in ultra-endurance disciplines are needed.The importance of hydration requirements in hot and humid conditions in ultra-endurance running needs to be kept in mind.

## Figures and Tables

**Figure 1 nutrients-13-01085-f001:**
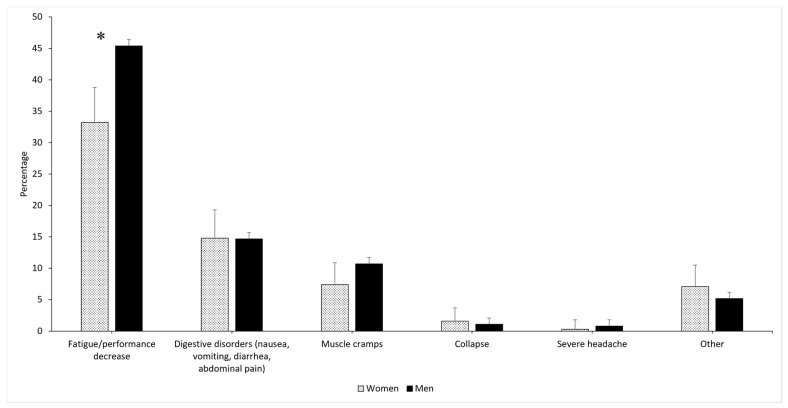
Heat-related symptoms experienced during the race by Grand Raid de la Réunion runners according to sex. * Significant difference between men and women.

**Figure 2 nutrients-13-01085-f002:**
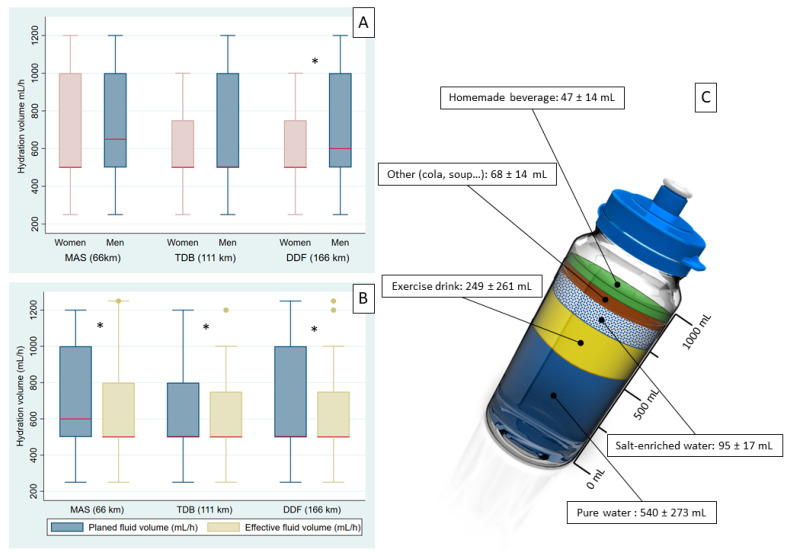
Hydration strategies of runners of the “Grand Raid de la Réunion” 2019 edition. MAS: *La Mascareignes*, TDB: *Le Trail de Bourbon*, DDF: *La Diagonale des Fous*. (**A**) Hydration volume in mL/h according to race and sex. * Significant difference between men and women. (**B**) Planned and effective hydration volume in mL/h reported by runners according to race. * Significant difference between planned and effective hydration volume. (**C**) Distribution of beverage types extrapolated on a 1-L bottle. Data are presented in milliliters (mL) with mean ± SD.

**Table 1 nutrients-13-01085-t001:** Characteristics of study participants in the Grand Raid de la Réunion runners by total sample and according to the race.

Variables	Total	*La Mascareignes*	*Le Trail De Bourbon*	*La Diagonale Des Fous*
**Demographic characteristics**				
Sex				
Men	2601 (83.2)	630 (74.5) ^#^	498 (78.7) ^$^	1468 (89.5) *
Women	525 (16.8)	216 (25.5)	135 (21.3)	173 (10.5) *
Age (years)	42 (35–50)	38 (32–46) ^#^	41 (33–50) ^$^	44 (38–50) *
Place of residence				
Réunion Island	1605 (52.0)	596 (71.0)	420 (66.9) ^$^	589 (36.4) *
Metropolitan France	1310 (42.5)	208 (24.8)	179 (28.5)	923 (57.0)
Other country	171 (5.5)	35 (4.2)	29 (4.6)	107 (6.6)
Living in a tropical climate	1688 (54.4)	611 (72.7)	440 (70.1) ^$^	635 (39.1) *
**Morphological characteristics**				
Men				
Mass (kg)	71 (66–76)	72 (66–77)	71 (66–76)	71 (66–75)
Height (cm)	176 (172–181)	176 (172–181)	176 (172–181)	176 (172–181)
BMI (kg/m2)	22.8 (21.5–24.2)	22.9 (21.5–24.3)	22.6 (21.4–24.2)	22.7 (21.5–24.0)
Women				
Mass (kg)	56 (52–61)	56 (52–62)	56 (52–62)	56 (52–60)
Height (cm)	165 (160–170)	165 (160–170)	165 (160–170)	164 (160–168)
BMI (kg/m2)	20.7 (19.6–22.0)	20.8 (19.5–22.2)	20.7 (19.3–22.2)	20.7 (19.8–21.7)
**Training characteristics**				
Trail running experience (yr)	5 (3–10)	3 (1–6) ^#^	5 (3–9) ^$^	6 (4–10) *
Number of ultra-races >60 km ran throughout career (*n*)	5 (2–10)	1 (0–3) ^#^	3 (2–7) ^$^	8 (4–13) *
Yearly number of ultra-races >60 km (*n*/yr)	1 (0.4–1.5)	0.1 (0–0.7) ^#^	0.9 (0.5–1.3) ^$^	1.3 (0.8–2.0) *
Average weekly training (over the 6-month period before the race)				
Duration (h)	8 (5–10)	6 (4–8) ^#^	7 (5–10) ^$^	8 (6–12) *
Distance (km)	50 (30–60)	40 (25–50) ^#^	40 (30–50) ^$^	50 (40–70) *
Ascent (m)	1200 (700–2000)	1000 (500–1500) ^#^	1000 (700–2000) ^$^	1500 (900–2000) *

Categorical variables are expressed as *n* (percentages) and quantitative variables as medians (Q1–Q3). Percentages are calculated on the number of respondents for each variable. BMI = Body Mass Index. * Comparisons between La Diagonale des Fous (DDF) vs. La Mascareignes (MAS), ^$^ comparisons between DDF vs. Le Trail de Bourbon (TDB), ^#^ comparisons between MAS vs. TDB. *, ^$^, ^#^ significant difference after Bonferroni correction.

**Table 2 nutrients-13-01085-t002:** Previous history of symptoms and diagnosis of heat illness in the Grand Raid de la Réunion runners by total sample and according to sex and climate of residence.

Variables	Total (*n* = 2286)	Women (*n* = 410)	Men (*n* = 1876)	Hot Climate (*n* = 1250)	Temperate Climate (*n* = 1033)
History of heat-related symptoms	1771 (78.2)	287 (70.2)	1484 (79.9) *	997 (80.5) *	771 (75.3)
Fatigue/performance decrease	1318 (58.2)	226 (55.3)	1092 (58.8)	723 (58.4)	593 (57.9)
Muscle cramps	852 (37.6)	92 (22.5)	760 (40.9) *	551 (44.5) *	298 (29.1)
Digestive disorders	361 (15.9)	55 (13.5)	306 (16.5)	170 (13.7) *	191 (18.7)
Severe headache	98 (4.3)	25 (6.1)	73 (3.9)	65 (5.3)	33 (3.2)
Collapse	61 (2.7)	12 (2.9)	49 (2.6)	29 (2.3)	32 (3.1)
Other	24 (1.1)	6 (1.5)	18 (1.0)	12 (1.0)	12 (1.2)
History of heat illness diagnosis	43 (1.9)	4 (1.0)	39 (2.1)	20 (1.6)	23 (2.2)
Dehydration	24 (1.1)	0	24 (1.3)	9 (0.7)	15 (1.5)
Hyponatremia	3 (0.1)	2 (0.5)	1 (0.05)	0	3 (0.3)
Heat exhaustion	6 (0.3)	0	6 (0.3)	3 (0.2)	3 (0.3)
Heat stroke	14 (0.6)	2 (0.5)	12 (0.6)	6 (0.5)	8 (0.8)
Other	6 (0.3)	0	6 (0.3)	4 (0.3)	2 (0.2)
Hospitalization	18 (0.8)	4 (1.0)	14 (0.8)	5 (0.4)	13 (1.3) *
Intensive Care Unit hospitalization	4 (0.2)	1 (0.2)	3 (0.2)	1 (0.1)	3 (0.3)

Data are expressed as *n* (percentages). All * refer to comparisons between (1) women vs. men, or (2) runners from hot climates vs. runners from temperate climates. Percentages are calculated on the number of respondents for each variable. ***** Significant difference after Bonferroni correction.

**Table 3 nutrients-13-01085-t003:** Planned hydration and per-cooling strategies in the Grand Raid de la Réunion runners by total sample and according to sex and climate of residence.

Variables	Total (*n* = 2286)	Women (*n* = 410)	Men (*n* = 1876)	Hot Climate (*n* = 1250)	Temperate Climate (*n* = 1033)
Planned cooling strategy	880 (45.1)	122 (40.8)	681 (45.9)	422 (45.8)	374 (44.4)
Stop/rest in the shade	379 (19.4)	54 (18.1)	292 (19.7)	200 (21.7)	146 (17.3)
Wet sponge	232 (11.9)	34 (11.4)	179 (12.1)	108 (11.7)	104 (12.4)
Leg immersion in cold water (creek, river)	131 (6.7)	25 (8.4)	96 (6.5)	69 (7.5)	50 (5.9)
Shower/whole body immersion	71 (3.6)	7 (2.3)	53 (3.6)	35 (3.8)	24 (2.9)
Head/neck cooling	327 (16.8)	52 (17.4)	252 (17.0)	136 (14.8)	164 (19.5)
Cooling of other body area	50 (2.3)	7 (2.3)	57 (3.8)	24 (2.6)	39 (4.6)
Cold towel	80 (4.1)	7 (2.3)	64 (4.3)	44 (4.8)	26 (3.1)
Hat, cap, etc.	239 (12.3)	36 (12.0)	184 (12.4)	86 (9.3)	131 (15.6) *
Neck collar	55 (2.8)	10 (3.3)	44 (3.0)	26 (2.8)	26 (3.1)
Ice slurry/water ingestion	33 (1.7)	6 (2.0)	24 (1.6)	24 (2.6) *	5 (0.6)
Ice vest	5 (0.3)	0	5 (0.3)	3 (0.3)	1 (0.1)
Other	23 (1.2)	2 (0.7)	18 (1.2)	12 (1.3)	8 (1.0)
Planned fluid consumption					
Pure Water	1688 (75.3)	299 (74.2)	1389 (75.5)	968 (78.4) *	720 (71.4)
Sodium-enriched water	701 (31.3)	96 (23.8)	605 (32.9) *	350 (28.3)	351 (34.8) *
Homemade preparation	306 (13.6)	66 (16.4)	240 (13.0)	207 (16.8) *	99 (9.8)
Exercise drink	1341 (59.8)	222 (55.1)	1119 (60.8)	733 (59.4)	608 (60.3)
Other	416 (18.6)	80 (19.9)	336 (18.3)	242 (19.6)	174 (17.3)
Pure water alone (no other beverage)	237 (10.6)	44 (10.9)	193 (10.5)	121 (9.8)	116 (11.5)

Data are expressed as *n* (percentages). All * refer to comparisons between (1) women vs. men, or (2) runners from hot climates vs. runners from temperate climates. Percentages are calculated on the number of respondents for each variable. ***** Significant difference after Bonferroni correction.

## Data Availability

The data presented in this study are available on reasonable request from the corresponding author.

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
