# Peer review of "Heat Acclimatization, Cooling Strategies, and Hydration during an Ultra-Trail in Warm and Humid Conditions"

_nutrients, 2021, doi:10.3390/nu13041085_

Round 1
Reviewer 1 Report
1. Thank you for asking me to review this paper. It is an interesting survey discussing runners' experience of heat illness symptoms and their mitigation by cooling techniques and fluid intake. The number of participants is very large (3,300 runners) and the authors are to be congratulated for this, especially given the difficulties of running field studies of this nature. 2. The introduction is a well-written review of the current discussions around fluid management, and the difficulties in extrapolating current marathon guidance (drink to thirst) to longer endurance events, and the need therefore for surveys of this nature and further studies. I felt that the discussion around EHI could be more detailed, especially around the definitions, as this is a principal part of the survey to follow. EHS (usually heatstroke not heat stress) is usually defined as a core temp greater than 40C with CNS abnormalities (Bouchama 2002). 3. The methodology and discussion around fluid intake is generally good but more clarity around what effective hydration signifies would be useful. The fluid intake strategies are documented in around 60% of runners - do the authors have details on the other? 4. Similarly the discussions around cooling strategies and heat acclimation are good. 5. My main concern with the paper is around the definitions used in the heat discussion. It would appear that while all of the symptoms in the HRS group might be associated with heat illness, all can also be present without heat illness, and using these to define HRS may not be robust. I don't have access to the questionnaires. Similarly, dehydration and hyponatraemia may not signify heat illness. I would be interested to know how the heat exhaustion and heat stroke were diagnosed, whether by clinical assessment (only around a third appear to have been treated in hospital) and again, diagnosing heat illness using the criteria presented here would not be robust. Refs 23 and 24 (heat illness in other ultra-endurance events) quote a core temperature consistent with EHI. Without this data in the current paper and just relying on participants' statements, especially without these being defined, would make this comparison difficult on the basis of what has been presented here. 6. There are a number of grammatical and spelling errors throughout the paper, and it would benefit from further proofreading.
Author Response
- Thank you for asking me to review this paper. It is an interesting survey discussing runners' experience of heat illness symptoms and their mitigation by cooling techniques and fluid intake. The number of participants is very large (3,300 runners) and the authors are to be congratulated for this, especially given the difficulties of running field studies of this nature.
The authors thank the reviewer for their positive comments. The changes are indicated in the text in red and our replies are included below.
- The introduction is a well-written review of the current discussions around fluid management, and the difficulties in extrapolating current marathon guidance (drink to thirst) to longer endurance events, and the need therefore for surveys of this nature and further studies. I felt that the discussion around EHI could be more detailed, especially around the definitions, as this is a principal part of the survey to follow. EHS (usually heatstroke not heat stress) is usually defined as a core temp greater than 40C with CNS abnormalities (Bouchama 2002).
In line with the reviewer comment, EHS has been defined based on an elevated core temperature and a CNS dysfunction. Regarding the core temperature value used in the definition, the authors recognize that previous papers have used 40ºC, but used 40.5ºC as it is becoming a more standard threshold within the latest publications (Epstein 2019). In addition, Exertional Heat Stress has been replaced by Exertional Heat Stroke (EHS) as this was a typo.
=> line 35-36:
“Its most severe form, exertional heat stroke (EHS) is characterized by a core body temperature above 40.5°C associated to central nervous system dysfunction (delirium, convulsions, coma)” [3,4]
- The methodology and discussion around fluid intake is generally good but more clarity around what effective hydration signifies would be useful.
A precision has been included in definition of “planed hydration” versus “effective hydration” in materials and methods section.
=> line 124-127
“Planned hydration was asked in pre-race survey by “How amount of fluid do you plan to drink during the race? (in mL/h)” and effective hydration was asked in post-race survey by “how amount of fluid did you drink on during the race (in mL/h)?” (Appendix A1 and A2).”
The fluid intake strategies are documented in around 60% of runners - do the authors have details on the other?
The fluid intake strategies were documented only on the pre-race survey that only 2,286 (69% of the total sample) runners had filled. Unfortunately, no data on the fluid intake of the other runners was available, excepted the effective fluid volume asked in the post-race survey (Figure 2.).
- Similarly the discussions around cooling strategies and heat acclimation are good.
The authors thank the reviewer for the positive comment.
- My main concern with the paper is around the definitions used in the heat discussion. It would appear that while all of the symptoms in the HRS group might be associated with heat illness, all can also be present without heat illness, and using these to define HRS may not be robust. I don't have access to the questionnaires. Similarly, dehydration and hyponatraemia may not signify heat illness. I would be interested to know how the heat exhaustion and heat stroke were diagnosed, whether by clinical assessment (only around a third appear to have been treated in hospital) and again, diagnosing heat illness using the criteria presented here would not be robust. Refs 23 and 24 (heat illness in other ultra-endurance events) quote a core temperature consistent with EHI. Without this data in the current paper and just relying on participants' statements, especially without these being defined, would make this comparison difficult on the basis of what has been presented here.
The authors thank the reviewer for this important remark which is one of the limits of this work: all HRS and HRI were self-assessed by the runners. Although it was specified in the questionnaires that each symptom, diagnosis, or history of interest had to be heat-related, the authors were subject to athlete’s declarative bias. No clinical information was available for those who reported HRI or HRS to confirm the diagnosis (core temperature for example). The authors agree with the reviewer that similar HRS can be present without heat illness. A new paragraph has been added in the "Study Strengths and Limitations" section to caution the reader about this limitation of our work.
=> line 416-424
“As this study is a survey-based study, it is subject to the declarative bias of athletes. It remains possible that despite the specify of the questions regarding HRS and HRI some runner attributed to heat some conditions being not heat-related. Comparisons of prevalence made with other disciplines or other papers should therefore be considered with caution due to the lack of clinical data in our study to confirm the self-assessment of the runners. However, our study provides a relevant estimation of the burden of EHI self-reported in ultra-endurance running, a discipline poorly investigated,”
Lastly, a new section “practical implication” was added where the importance of clinical assessment of EHI (core temperature, symptoms) to better estimate the burden of heat stress in ultra-endurance disciplines has been developed.
Line 453-463
.Practical implications
- Need to inform ultra-endurance runners about benefits of heat acclimatization and cooling strategies before running in hot and humid conditions.
- Prevalence of EHI remains low in ultra-endurance based on declarative evaluation in this study. However, prospective study with clinical assessment of EHI (core temperature, symptoms) to better estimate the burden of heat stress in ultra-endurance disciplines are needed.
- Keep on mind the importance of hydration requirements in hot and humid conditions in ultra-endurance running.
- There are a number of grammatical and spelling errors throughout the paper, and it would benefit from further proofreading.
The manuscript has been carefully proofread and various spelling mistakes have been addressed (as highlighted in red in the text).
Reviewer 2 Report
Thank you for allowing me to read this manuscript. I found this to be an interesting study and well-written. As such, I only have some minor comments that the authors may wish to consider. I feel that it would be helpful to include working definitions e.g. a w definition of a ‘cooling strategy’ would be useful, and did the participants understand it in a similar manner? It would also be useful to consider a more explicit practical implications section in the discussion.
Abstract: The conclusions reads more like a practical implications section rather than a conclusions. Whilst I appreciate that this is useful advice, I feel that the conclusion should be more clearly related to the presented data.
Introduction:
Line 31: Defining hot and humid ambient conditions would be a useful addition. The same could be applied to exertional heat stress as well (line 32)
The information provided is interesting, but could be used to provide a stronger rationale for the study.
Methods:
Table 1: Consider reporting age as an integer.
Results:
Figure 2C: This is an interesting way of presenting data. Is there any way to add a scale in order to provide context for the different proportions?
Discussion: Is there any merit analysing the 61 non-finishers to examine if they differed from the finishers?
A specific section relating to the practical implications would be beneficial.
Author Response
Thank you for allowing me to read this manuscript. I found this to be an interesting study and well-written. As such, I only have some minor comments that the authors may wish to consider.
The authors thank the reviewer for their positive comments. The changes are indicated in the text in red and our replies are included below.
I feel that it would be helpful to include working definitions e.g. a w definition of a ‘cooling strategy’ would be useful, and did the participants understand it in a similar manner?
A definition of cooling strategy was added in the Materials and Methods section (survey paragraph). Explanations of the main cooling methods were available in the survey provided to the runners. It was included to the manuscript as follows:
“Cooling strategy was defined as any method used for reducing or prevent excessive heat storage during exercise. Only mid-cooling (i.e., cooling during the races) strategies used by runners were assessed. To help the runners answering, the main cooling methods were provided in the questionnaire (Appendix A1).” Lines 118-121.
It would also be useful to consider a more explicit practical implications section in the discussion.
The authors thank the reviewer for this advice. We have added a practical implication section.
Line 453-463
.Practical implications
- Need to inform ultra-endurance runners about benefits of heat acclimatization and cooling strategies before running in hot and humid conditions.
- Prevalence of EHI remains low in ultra-endurance based on declarative evaluation in this study. However, prospective study with clinical assessment of EHI (core temperature, symptoms) to better estimate the burden of heat stress in ultra-endurance disciplines are needed.
- Keep on mind the importance of hydration requirements in hot and humid conditions in ultra-endurance running.
Abstract: The conclusions reads more like a practical implications section rather than a conclusions. Whilst I appreciate that this is useful advice, I feel that the conclusion should be more clearly related to the presented data.
The abstract has been amended in line with the reviewer concerns. It now reads:
“The present study shows that ultra-endurance runners have a wide variability of hydration and heat preparation strategies. Understanding of heat stress repercussion in ultra-endurance running need to be improved by specific on field research”. Lines 25-27.
Introduction:
Line 31: Defining hot and humid ambient conditions would be a useful addition. The same could be applied to exertional heat stress as well (line 32). The information provided is interesting, but could be used to provide a stronger rationale for the study.
Extertional Heat Stroke has been defined in the revised manuscript. Regarding the conditions impairing performance, the authors prefer to do not give a specific threshold as this is a continuum with some studies reporting alterations in Marathon from as low as 11ºC, but there is a lack of data in ultra-endurance. Moreover, performance depend on temperature, humidity, wind speed and radiation, and it would not be accurate to give a definition based on a single parameter.
Methods:
Table 1: Consider reporting age as an integer.
Age is now reported as an integer (Table 1).
Results:
Figure 2C: This is an interesting way of presenting data. Is there any way to add a scale in order to provide context for the different proportions?
A graduation was added on the bottle. The direct information of proportion is precisely detailed for each beverage type (mean±SD) (Figure 2C).
Discussion: Is there any merit analysing the 61 non-finishers to examine if they differed from the finishers?
Specifics statistics are presented for non-finishers compared to finishers for Hydration rate (line196) and arrivals time before the race (line 302). The authors chose not to go further in the comparisons because of the low response rate of non-finishers (3% in the study) compared to the true withdraw rate during the race (28%), which would have necessarily led to bias in our analysis.
A specific section relating to the practical implications would be beneficial.
Amended Line 453-463
Reviewer 3 Report
An interesting, informative and well written manuscript: however, there are some writing/editing issues that the authors should consider and address. In the Abstract, line 13, "...to assess the history of exertional ...". Line 17, "...and 74+/_17%, respectively. Line 26, "...prolonged exercises, but partially ...". In the Introduction, line 39, "different compared to shorter exercises ....". Line 65 "...ultra-endurance [20,21], hydration guidelines ...". Line 66, "...endurance exercise and use of ...". In the Results section, lines 132-133, "...higher prevalence of a history of ...". Line 150, "...vs 5[3-6], respectively, ...". Line 152, "...45.1% of runners with no ...". In the Heat related symptoms during the races section, lines 5 & 6, "...their performance, without reaching the level ..". Line 7, "Up to 54.6% of the runners reported ...". In the Hydration section, paragraph 1, line 6, "...finishers (p=0.009). Line 15, "...illustrated in Figure 2C. Qualitative analysis .....". Paragraph 2, line 10, "...men (73.9% vs 81.5%, respectively, ...". Line 12, "Moreover, 58.8% of the runners ....". In the Discussion section, in the History of heat-related symptoms and Exertional heat illness section, line 1, "Three-quarters of our ...". Line 9, "...epidemiological studies, but can ...". Lines 15 & 16, "...physiological damage); and therefore, be more ...". Line 17, "Finally, a relatively low prevalence of ...". In the Heat training and acclimatization section, line 16, "First, this may be due to a lack ...". Line 21, "...In our study, 42% of the runners who had ...". In the Heat related symptoms during the races section, line 5, "...with temperatures not exceeding ...". Line 6, "...remaining at usual levels ...". Line 14, "Up to 55% of the runners reported ...". Line 15, "...that only 20% of the runners ...". Lines 17 & 18, "...fact that respondents reporting symptoms were not due ....". Line 21, "considered that it did not limit ...". Line 33, "...before the start of the hot season ...". Lines 33 & 34, "...for local residents (mean temperature ...". Line 35, "Although temperatures in France were ...". Line 38, "...may not confer an advantage if one has ....". Line 40, "acclimatization of a hot season as was the case from the runners ...". Line 41, "union, i.e. the vast ...". Line 43, "...shown that the risk of heat ...". Line 46, "...recent study of the analysis of heat ...". Line 48, "...in Asia in the winter when ...". In the Hydration section, paragraph 1, line 4, "...in shorter races: 686+/_237 ...". Line 6, "These results are in line with ...". Line 16, "...should be cautious to avoid ...". Line 20, "conditions can increase the risk ...". Line 23, "...an acute kidney insult in ultra-marathon ...". In the Conclusions section, lines 6 & 7, "...than environmental parameters. Being native as a resident of a tropical country ...".
Author Response
The authors thank the reviewer for their positive comments and the in-depth proofreading with advice of writing/editing issues. All suggestions for changes have been made in the text in red.
Round 2
Reviewer 1 Report
Thank you for asking me to review this paper and for accommodating my suggestions. The discussion on hydration and cooling strategies are good, but I would still have concerns about the reporting of heat-related symptoms:
I understand that the subjects were asked to report the symptoms in table 2 (fatigue / muscle cramps / digestive disorders / headache / collapse / dehydration / hyponatremia / heat stroke) but without any clinical assessment. I have two concerns: (1) I would question how patients can diagnose hyponatraemia and heatstroke themselves, and (2) the majority of these symptoms can occur without heat illness and could therefore not be used to diagnose heat-related illness. Unfortunately I don't think the heat-related section is robust.
Author Response
The reviewer is right that the athletes are not qualified to self-diagnose hyponatremia and heatstroke. However, we differentiated two different parts in the questionnaire and in this article the first part was questions about the runners' medical history before this race in the pre-race questionnaire.
For more precision we added in manuscript "medical history of heat illness before this race" (line 140-141) in the results section. In the methodology we also specified that "medical history of heat symptoms and heat illness before this race were collected " (line 117) in the survey section.
2) The second part concerns the declaration of the incidence of heat-related symptoms during this race. Question was in the post-race questionnaire: "During your race, did you experience any of the following symptoms, which you think were related to heat or poor hydration management (Multiple answers possible)?" The proposals here were only related to symptoms: Nausea, Vomiting, Cramps, Fatigue/diminution of performance, Severe headaches, Malaise/loss of consciousness, None of the above. Concerning these symptoms experienced during the race, we did not ask the runners about exercise hyponatremia associated or exercise heat stroke, which, as mentioned by the reviewer, require the intervention of a physician to make the diagnosis and clinical outcomes (central temperature measurement, biological measurements, etc.).
The discussion in our article distinguishes between these two parts:
section 1 of discussion : “History of heat-related symptoms and Exertional heat illness” we amended by “History of heat-related symptoms and Exertional heat illnes before this race”.
section 5 of discussion : Heat related symptoms during the races
This section is only concerned with symptoms.
In start of discussion section, we amended “the main results of the present study are that: i) although 78% of ultra-endurance runners had a previous history of HRS, 1.9% declared a medical history of EHI…” line 235-236.
The pre and post-race surveys (in french) are available in attachment for more understanding.
